# Explanation by Progressive Exaggeration

**Sumedha Singla**
Department of Computer Science
University of Pittsburgh

**Brian Pollack, Junxiang Chen**
Department of Biomedical Informatics
University of Pittsburgh

**Kayhan Batmanghelich**
Department of Biomedical Informatics
Department of Computer Science
Intelligent Systems Program
University of Pittsburgh

## Abstract

As machine learning methods see greater adoption and implementation in high stakes applications such as medical image diagnosis, the need for model interpretability and explanation has become more critical. Classical approaches that assess feature importance (*e.g.*, saliency maps) do not explain *how* and *why* a particular region of an image is relevant to the prediction. We propose a method that explains the outcome of a classification black-box by gradually *exaggerating* the semantic effect of a given class. Given a query input to a classifier, our method produces a progressive set of plausible variations of that query, which gradually changes the posterior probability from its original class to its negation. These counter-factually generated samples preserve features unrelated to the classification decision, such that a user can employ our method as a "tuning knob" to traverse a data manifold while crossing the decision boundary. Our method is model agnostic and only requires the output value and gradient of the predictor with respect to its input.

## 1 Introduction

With the explosive adoption of deep learning for real-world applications, explanation and model interpretability have received substantial attention from the research community (Kim, 2015; Doshi-Velez & Kim, 2017; Molnar, 2019; Guidotti et al., 2019). Explaining an outcome of a model in high stake applications, such as medical diagnosis from radiology images, is of paramount importance to detect hidden biases in data (Cramer et al., 2018), evaluate the fairness of the model (Doshi-Velez & Kim, 2017), and build trust in the system (Glass et al., 2008). For example, consider evaluating a computer-aided diagnosis of Alzheimer's disease from medical images. The physician should be able to assess whether or not the model pays attention to age-related or disease-related variations in an image in order to trust the system. Given a query, our model provides an explanation that gradually *exaggerates* the semantic effect of one class, which is equivalent to traversing the decision boundary from side to another.

Although not always clear, there are subtle differences between interpretability and *explanation* (Turner, 2016). While the former mainly focuses on building or approximating models that are locally or globally interpretable (Ribeiro et al., 2016), the latter aims at explaining a predictor a-posteriori. The explanation approach does not compromise the prediction performance. However, a rigorous definition for what is a good explanation is elusive. Some researchers focused on providing feature importance (*e.g.*, in the form of a heatmap (Selvaraju et al., 2017)) that influence the outcome of the predictor. In some applications (*e.g.*, diagnosis with medical images) the causal changes are spread out across a large number of features (*i.e.*, large portions of the image are impacted by a disease). Therefore, a heatmap may not be informative or useful, as almost all image features are highlighted. Furthermore, those methods do not explain *why* a predictor returns an outcome. Others have introduced local occlusion or perturbations to the input (Zhou et al., 2014; Fong & Vedaldi, 2017) by assessing which manipulations have the largest impact on the predictors. There is also

recent interest in generating counterfactual inputs that would change the black box classification decision with respect to the query inputs (Goyal et al., 2019; Liu et al., 2019). Local perturbations of a query are not guaranteed to generate realistic or plausible inputs, which diminishes the usefulness of the explanation, especially for end users (*e.g.*, physicians). We argue that the explanation should depend not only on the predictor function but also on the data. Therefore, it is reasonable to train a model that learns from data as well as the black-box classifier (*e.g.*, (Chang et al., 2019; Dabkowski & Gal, 2017; Fong & Vedaldi, 2017)).

Our proposed method falls into the local explanation paradigm. Our approach is model agnostic and only requires access to the predictor values and its gradient with respect to the input. Given a query input to a black-box, we aim at explaining the outcome by providing *plausible* and *progressive* variations to the query that can result in a change to the output. The plausibility property ensures that perturbation is natural-looking. A user can employ our method as a "tuning knob" to progressively transform inputs, traverse the decision boundary from one side to the other, and gain understanding about how the predictor makes a decision. We introduce three principles for an explanation function that can be used beyond our application of interest. We evaluate our method on a set of benchmarks as well as real medical imaging data. Our experiments show that the counterfactually generated samples are realistic-looking and in the real medical application, satisfy the external evaluation. We also show that the method can be used to detect bias in training of the predictor.

## 2 METHOD

Consider a *black box* classifier that maps an input space $\mathcal{X}$ (*e.g.*, images) to an output space $\mathcal{Y}$ (*e.g.*, labels). In this paper, we consider binary classification problems where $\mathcal{Y} = \{-1, +1\}$. To model the black-box, we use $f(\mathbf{x}) = \mathbb{P}(y|\mathbf{x})$ to denote the posterior probability of the classification. We assume that $f$ is a differentiable function and we have access to its value as well as its gradient with respect to the input $\nabla_{\mathbf{x}} f(\mathbf{x})$.

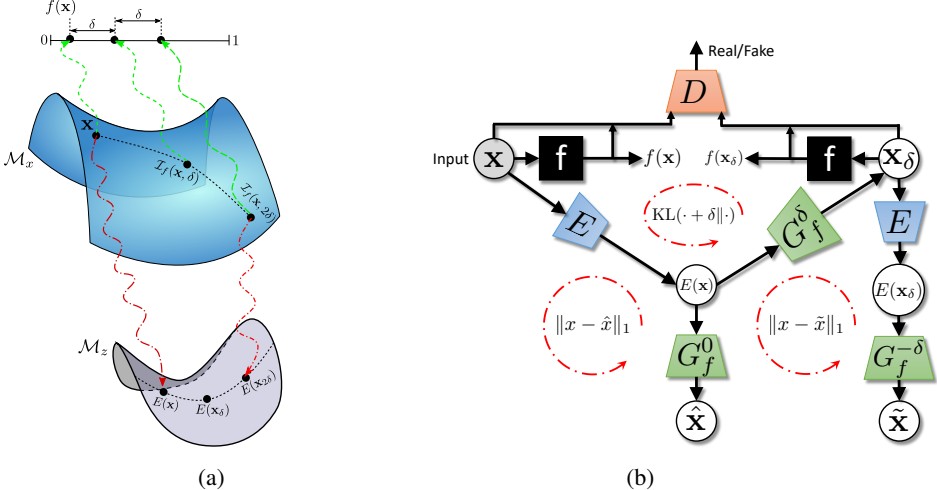

(a)  (b)

Figure 1: **(a)** The schematic of the method: $f$ is the black-box function producing the posterior probability. $\delta$ is the required change in black-box's output $f(\mathbf{x})$. $\mathcal{I}_f(\mathbf{x}, \delta)$ is an explainer function for $f$, which shifts the value of $f(\mathbf{x})$ by $\delta$. The $E(\cdot)$ is an encoder that maps the data manifold $\mathcal{M}_x$ to the embedding manifold $\mathcal{M}_z$. $\mathbf{x}_\delta$ is an abbreviation for $\mathcal{I}_f(\mathbf{x}, \delta)$. **(b)** The architecture of our model: $E$ is the encoder, $G_f^\delta$ denotes the conditional generator $G(\cdot, c_f(x, \delta))$, $f$ is the black-box and $D$ is the discriminator. The circles denote loss functions.

We view the (visual) explanation of the black-box as a generative process that produces an input for the black-box that slightly perturbs current prediction ($f(\mathbf{x}) + \delta$) while remaining plausible and realistic. By repeating this process towards each end of the binary classification spectrum, we can traverse the prediction space from one end to the other and exaggerate the underlying effect. We conceptualize the traversal from one side of the decision boundary to the other as walking across a data manifold, $\mathcal{M}_x$. We assume the walk has a fixed step size and each step of the walk makes $\delta$ change to the posterior probability of the the classifier, $f$. Since the output of $f$ is bounded between

$[0, 1]$, we can take at-most $\lfloor \frac{1}{\delta} \rfloor$ steps. Each positive (negative) step increases (decreases) the posterior probability of the previous step. We assume that there is a low-dimensional embedding space ($\mathcal{M}_z$) that encodes the walk. An encoder, $E : \mathcal{M}_x \to \mathcal{M}_z$, maps an input, $\mathbf{x}$, from the data manifold, $\mathcal{M}_x$, to the embedding space. A generator, $G : \mathcal{M}_z \to \mathcal{M}_y$, takes both the embedding coordinate and the number of steps and maps it back to the data manifold (see Figure 1).

We use $\mathcal{I}_f(\cdot, \cdot)$ to denote the explainer function. Formally, $\mathcal{I}_f(\mathbf{x}, \delta) : (\mathcal{X}, \mathbb{R}) \to \mathcal{X}$ is a function that takes two arguments: a query image $\mathbf{x}$ and the desired perturbation $\delta$. This function generates a perturbed image which is then passed through function $f$. The difference between the outputs of $f$ given the original image and the perturbed image should be the desired change *i.e.*, $f(\mathbf{x}_\delta) - f(\mathbf{x}) = \delta$. We use $\mathbf{x}_\delta$ to denote $\mathcal{I}_f(\mathbf{x}, \delta)$. This formulation enables us to use $\delta$ as a knob to *exaggerate* the visual explanations of the query sample while it is crossing the decision boundary given by function $f$. Our proposed interpretability function $\mathcal{I}_f$ should satisfy the following properties:

1. **Data Consistency:** perturbed samples generated by $\mathcal{I}_f$ should lie on the data manifold, $\mathcal{M}_x$, to be consistent with real data. In other words, the generated samples should look realistic when compared to other samples.

2. **Compatibility with $f$:** changing the second argument in $\mathcal{I}_f(\mathbf{x}, \cdot)$ should produce the desired outcome from classifier $f$, *i.e.*, $f(\mathcal{I}_f(\mathbf{x}, \delta)) \approx f(\mathbf{x}) + \delta$.

3. **Self Consistency:** Applying reverse perturbation should bring $\mathbf{x}$ back to its original form *i.e.*, $\mathcal{I}_f(\mathcal{I}_f(\mathbf{x}, \delta), -\delta) = \mathbf{x}$. Also, applying setting $\delta$ to zero should return the query, *i.e.*, $\mathcal{I}_f(\mathbf{x}, 0) = \mathbf{x}$.

Each criterion is enforced via a loss function which are discussed in the following sections.

## 2.1 DATA CONSISTENCY

We adopt the Generative Adversarial Networks (GANs) framework for our model (Goodfellow et al., 2014). The GANs implicitly model the underlying data distribution by setting up a min-max game between generative ($G$) and discriminative ($D$) networks:

$$\mathcal{L}_{\text{GAN}}(D, G) = \mathbb{E}_{\mathbf{x}, c \sim P(\mathbf{x})}\big[\log\big(D(\mathbf{x})\big)\big] + \mathbb{E}_{\mathbf{z} \sim P_{\mathbf{z}}}\big[\log\big(1 - D(G(\mathbf{z}))\big)\big],$$

where $\mathbf{z}$ and $P_{\mathbf{z}}$ are the noise distribution and the corresponding canonical distribution. There has been significant progress toward improving GANs stability as well as sample quality (Brock et al., 2019; Karras et al., 2019). The advantage of GANs is that they produce realistic-looking samples without an explicit likelihood assumption about the underlying probability distribution. This property is appealing for our application.

Furthermore, we need to provide the desired amount of perturbation to the black-box, $f$. Hence, we use a Conditional GAN (cGAN) that allows the incorporation of a context as a condition to the GAN (Mirza & Osindero, 2014; Miyato & Koyama, 2018). To define the condition, we fix the step size, $\delta$, and descritize the walk which effectively cuts the posterior probability range of the predictor (*i.e.*, $[0, 1]$) into $\lfloor \frac{1}{\delta} \rfloor$ equally-sized bins. Hence, one can view the perturbation from $f(\mathbf{x})$ to $f(\mathbf{x}) + \delta$ as changing the bin index from the current value $c_f(\mathbf{x}, 0)$ to $c_f(\mathbf{x}, \delta)$ where $c_f(\mathbf{x}, \delta)$ returns the bin index of $f(\mathbf{x}) + \delta$. We use $c_f(\mathbf{x}, \delta)$ as a condition to the cGAN.

The cGAN optimizes the following loss function:

$$\mathcal{L}_{\text{cGAN}}(D, G) = \mathbb{E}_{\mathbf{x}, c \sim P(\mathbf{x}, c)}\big[\log\big(D(\mathbf{x}, c)\big)\big] + \mathbb{E}_{\mathbf{z} \sim P_{\mathbf{z}}, c \sim P_c}\big[\log\big(1 - D(G(\mathbf{z}, c), c)\big)\big], \quad (1)$$

where $c$ denotes a condition. Instead of generating random samples from $P_{\mathbf{z}}$, we use the output of an encoder, $E(\mathbf{x})$, as input to the generator. Finally, the explainer function is defined as:

$$\mathcal{I}_f(\mathbf{x}, \delta) = G(E(\mathbf{x}), c_f(\mathbf{x}, \delta)). \quad (2)$$

Our architecture is based on Projection GAN (Miyato & Koyama, 2018), a modification of cGAN. An advantage of the Projection GAN is that it scales well with the number of classes allowing $\delta \to 0$. The Projection GAN imposes the following structure on the discriminator loss function:

$$\mathcal{L}_{\text{cGAN}}(D, \hat{G})(\mathbf{x}, \mathbf{c}) = \log \frac{p_{\text{data}}(\mathbf{c}|\mathbf{x})}{q(\mathbf{c}|\mathbf{x})} + \log \frac{p_{\text{data}}(\mathbf{x})}{q(\mathbf{x})} := r(c|\mathbf{x}) + \psi(\boldsymbol{\phi}(\hat{G}(\mathbf{z}))), \quad (3)$$

where $\mathcal{L}_{\text{cGAN}}(D, \hat{G})$ indicates the loss function in Eq. 1 when $\hat{G}$ is fixed, $\boldsymbol{\phi}(\cdot)$ and $\psi(\cdot)$ are networks producing vector (feature) and scalar outputs respectively. The $r(c|\mathbf{x})$ is a conditional ratio function which will be discussed in Section 2.2.

## 2.2 COMPATIBILITY WITH THE BLACK BOX

In our model, the condition $c$ is an ordered variable *i.e.*, $c_f(\mathbf{x}, \delta_1) < c_f(\mathbf{x}, \delta_2)$ when $\delta_1 < \delta_2$. Therefore, we adapt the first term in Eq. 3 to account for ordinal multi-class regression by transforming $c_f(\mathbf{x}, \delta)$ into $\lfloor \frac{1}{\delta} \rfloor - 1$ binary classification terms (Frank & Hall, 2001):

$$r(c = k | \mathbf{x}) := \sum_{i < k} \mathbf{v}_i^T \boldsymbol{\phi}(\mathbf{x}), \tag{4}$$

where $\boldsymbol{\phi}(\cdot)$ is the feature network in Eq. 3 and $\mathbf{v}_i$'s are parameters. We also need to ensure that plugging $\mathbf{x}_\delta$ into $f(\cdot)$ yields $f(\mathbf{x}) + \delta$ (*i.e.*, compatible with $f$). This condition is enforce by a KullbackLeibler (KL) divergence loss term. Adding the KL loss and the conditional ratio function we arrive at the following loss:

$$\mathcal{L}_f(D, G) := r(c|\mathbf{x}) + D_{\mathrm{KL}}\left(f(\mathbf{x}) + \delta \| f(\mathcal{I}_f(\mathbf{x}, \delta))\right).$$

While the first term is a function of both $G$ and $D$, the second term influences only the generator $G$.

## 2.3 SELF CONSISTENCY

We use a reconstruction loss term to enforce encoder-decoder consistency and satisfy the identity constraint of $\mathbf{x} = \mathcal{I}_f(\mathbf{x}, 0)$,

$$\mathcal{L}_{\mathrm{rec}}(G) = ||\mathbf{x} - G\left(E(\mathbf{x}), c_f(\mathbf{x}, 0)\right)||_1, \tag{5}$$

We also require that the perturbation is reversible (*i.e.*, $\mathcal{I}_f(\mathcal{I}_f(\mathbf{x}, \delta), -\delta) = \mathbf{x}$). We use a cycle-consistency (Zhu et al., 2017) loss to reconstruct the input from its corresponding perturbed image,

$$\mathcal{L}_{\mathrm{cyc}}(G) = ||\mathbf{x} - G(E(\mathbf{x}_\delta), c_f(\mathbf{x}, 0))||_1. \tag{6}$$

Note that the conditions for the generators in Eq. 5 and 6 are the same. However, in the former, we are reconstructing the input $\mathbf{x}$ from its latent space, but in the latter, we perturb $\mathbf{x}_\delta$ from the bin index $c_f(\mathbf{x}, \delta)$ back to original bin index $c_f(\mathbf{x}, 0)$.

## 2.4 OBJECTIVE FUNCTIONS

We adapted the hinge version of the adversarial loss for $\mathcal{L}_{\mathrm{cGAN}}(G, D)$.

$$\begin{aligned}\mathcal{L}_{\mathrm{cGAN}}(D) = &-\mathbb{E}_{\mathbf{x} \sim p_{\mathrm{data}}}\big[\min(0, -1 + D(\mathbf{x}, c_f(\mathbf{x}, 0)))\big] \\ &- \mathbb{E}_{\mathbf{x} \sim p_{\mathrm{data}}, c_f(\mathbf{x}, \delta) \in [0, \frac{1}{\delta}]}\big[\min(0, -1 - D(G(E(\mathbf{x}), c_f(\mathbf{x}, \delta)), c_f(\mathbf{x}, \delta)))\big]\end{aligned} \tag{7}$$

$$\mathcal{L}_{\mathrm{cGAN}}(G, E) = -\mathbb{E}_{\mathbf{x} \sim p_{\mathrm{data}}, c_f(\mathbf{x}, \delta) \in [0, \frac{1}{\delta}]}\big[D(G(E(\mathbf{x}), c_f(\mathbf{x}, \delta)), c_f(\mathbf{x}, \delta))\big] \tag{8}$$

The overall objective function is

$$\min_{E, G} \max_{D} \lambda_{\mathrm{cGAN}} \mathcal{L}_{\mathrm{cGAN}}(D, G) + \lambda_f \mathcal{L}_f(D, G) + \lambda_{\mathrm{rec}} \mathcal{L}_{\mathrm{rec}}(G) + \lambda_{\mathrm{rec}} \mathcal{L}_{\mathrm{cyc}}(G) \tag{9}$$

where $\lambda_{\mathrm{cGAN}}, \lambda_f, \lambda_{\mathrm{rec}}$ are the hyper-parameters that balance the importance of the loss terms.

## 3 RELATED WORK

Our work broadly relates to literature in interpretation methods that are designed to provide a visual explanation of the decisions made by a black-box function $f$, for a given query sample $\mathbf{x}$.

**Perturbation-based methods:** These methods provide interpretation by showing what minimal changes are required in $\mathbf{x}$ to induce a desirable output of $f$. Some methods employed image manipulation via the removal of image patches (Zhou et al., 2014) or the occlusion of image regions (Zhou et al., 2014) to change the classification score. Recently, the use of influence function, as proposed by (Koh & Liang, 2017) are applied as a form of data perturbation to modify a classifier's response. The authors in (Fong & Vedaldi, 2017) proposed the use of optimal perturbation, defined as removing the smallest possible image region in $\mathbf{x}$ that results in the maximum drop in classification score. In another approach, (Chang et al., 2019) proposed a generative process to find and fill the image

regions that correspond to the largest change in the decision output of a classifier. To switch the decision of a classifier, (Goyal et al., 2019) suggested generating counterfactuals by replacing the regions of **x** with patches from images with a different class label. All of the aforementioned works perform pixel- or patch-level manipulation to **x**, which may not result in natural-looking images. In contrast, our model enforces that the perturbed data be consistent with the unperturbed data to ensure that the perturbation is plausible. Furthermore, our method can be applied to general data and is not restricted to the imaging domain.

**Saliency map-based methods:** Saliency maps explains the decision of $f$ on **x** by highlighting the relevant regions of $x$. Some earlier work in this direction(Simonyan et al., 2013; Springenberg et al., 2015; Bach et al., 2015) focuses on computing the gradient of the target class with respect to **x** and considers the image regions with large gradients as most informative. Building on this work, the class activation map (CAM) (Zhou et al., 2016) and its generalized version Grad-CAM Selvaraju et al. (2017) and other variants such as LPR (Bach et al., 2015) use a linear or non-linear combination of the activation layers to derive relevance score for every pixel in an image. These gradient-based methods are not model-agnostic and require access to intermediate layers. Recently, Adebayo et al. (2018) have shown that some saliency methods are independent both of the model and of the data generating process. We used their propose evaluation to validate our interpretation model. The saliency maps are also prone to adversarial attacks as shown by Ghorbani et al. (2019) and Kindermans et al. (2017). Furthermore, if the causal effect of a class is distributed across an image, which is the case in radiology images, the saliency approaches highlight large sections of the image, which greatly reduce the usefulness of the interpretation.

**Generative explanation-based methods:** These are interpretation models that uses a generative process to produce visual explanations. The contrastive explanations method (CEM) (Dhurandhar et al., 2018) generates explanations that show minimum regions in **x** which must be present/absent for a particular classification decision. In another work, (Liu et al., 2019; Joshi et al., 2019; Samangouei et al., 2018) generates explanations that highlight what features should be changed in **x** so that the classifier confidence in the prediction is strengthen (prototype) or weakened (counterfactual). Our approach is aligned with these latter lines of work, although our method and model architecture is different. Our method allows for the gradual change of the class effect, and our consistency criteria result in high-quality feasible perturbation in **x**. We rigorously evaluate our method on real medical imaging applications, in addition to the curated computer vision datasets.

# 4 EXPERIMENTS

We set up four experiments to evaluate our method. First, we assess if our method satisfies the three criteria of the explainer function introduced in Section 2. We report both qualitative and quantitative results. Second, we apply our method on a medical image diagnosis task. We use external domain knowledge about the disease to perform a quantitative evaluation of the explanation. Third, we train two classifiers on biased and unbiased data and examine the performance of our method in identifying the bias. While our method does not produce a saliency map, in our last experiment, we use the two counterfactual samples on the boundary $[0, 1]$ to generate a saliency map and compare it with the other methods. In Appendix A, we show further experiments to evaluate our model in human experiments, to demonstrate its compatibility with a multi-label classifier and, an ablation study, to show the relative importance of each of the three criteria of the explainer function.

Our experiments are conducted on the CelebA (Liu et al., 2015) and CheXpert (Irvin et al., 2019) datasets. CelebA contains 200K celebrity face images, each with forty attribute labels. We considered binary classifier trained on the "smiling" and "young" attributes. CheXpert is a medical dataset containing 224K chest x-ray images from 65K patients and has labels for fourteen radio-graphic observations. We considered Cardiomegaly as the target class for generating explanations. All images are re-sized to $128 \times 128$ before processing.

## 4.1 EVALUATING THE CRITERIA OF THE EXPLAINER

Figure 2 reports the qualitative results on three datasets. Given a query image **x** at inference time, our model generates a series of images $\mathbf{x}_\delta$ as visual explanations, which gradually increase the posterior probability $f(\mathbf{x}_\delta)$ (top label). We show results for three prediction tasks: smiling or not-smiling,

young or old, and Cardiomegaly or healthy. The values on the top of each figure report the $f(\mathbf{x}_\delta)$'s. For Cardiomegaly, we show the outlines of the heart as well as its normalized size (values inside the parenthesis), which is indicative of the disease.

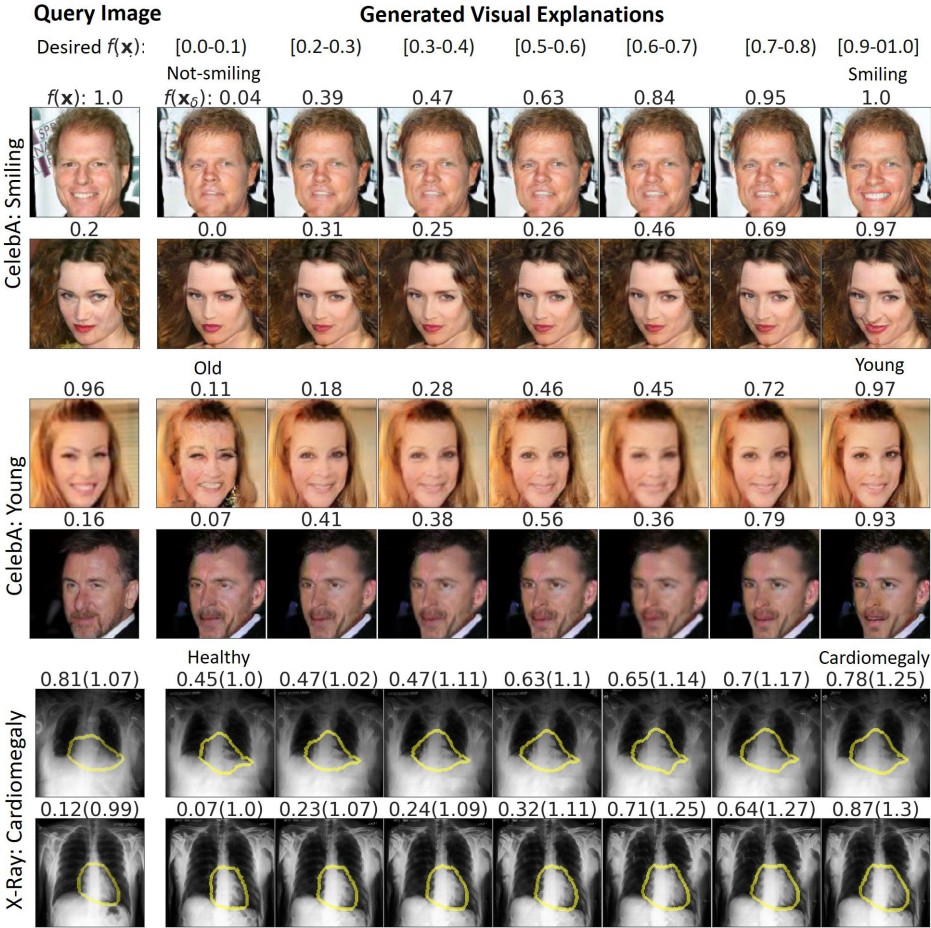

Figure 2: Visual explanations generated for three prediction tasks: smiling/not-smiling face (first two rows), young/old face (middle two rows) and Cardiomegaly/healthy chest x-ray (bottom two rows). The first column shows the query image, followed by the corresponding generated explanations. The values above each image are the output of the classifier $f$. For Cardiomegaly, we show the segmentation of the heart (yellow edge) and report normalized heart size (values in parenthesis), which is indicative of the disease.

**Data Consistency:** The generated explanations are synthesized variations of the query image. To quantitatively compare their visual quality, we consider Fréchet Inception Distance (FID) (Heusel et al., 2017). We compared our results against the counterfactual explanations produced by xGEM (Joshi et al., 2018). The details of the xGEM model are given in appendix A.2. We divided the real and fake (*i.e.*, generated explanations) images into two groups (on either boundary of $f(\mathbf{x}) \in [0, 1]$) and reported the FID for each group and the overall score. Our method significantly outperforms xGEM, producing crisper and more realistic-looking images. xGEM is based on variational autoencoder (VAE) which are known to produce blurry images (*see* Figure 7).

**Compatibility with the black-box $f$:** To quantify whether the generation process is aligned with the desire perturbation $\delta$, we plotted the expected outcome $f(\mathbf{x}) + \delta$ against the actual response of the classifier for the generated explanations, $f(\mathbf{x}_\delta)$. Figure 3 shows how our model performs when generating a series of explanations starting from a wide range of initial query images. The performance is almost perfect for Young/Old, but less so for more challenging classification problems such as Smiling or Cardiomegaly. The plot also validates that we are producing perturb images covering the entire classification range, $[0, 1]$. Appendix A.3 shows additional result from CelebA dataset.

| Target Class | CelebA:Smiling | | CelebA:Young | | Xray:Cardiomegaly | |
|---|---|---|---|---|---|---|
| | xGEM | Ours | xGEM | Ours | xGEM | Ours |
| Present ($f(\mathbf{x}_\delta) \in [0.9, 1]$) | 111.0 | **46.9** | 115.2 | **67.6** | 368.6 | **82.9** |
| Absent ($f(\mathbf{x}_\delta) \in [0, 0.1]$) | 112.9 | **56.3** | 170.3 | **74.4** | 394.6 | **84.8** |
| Overall ($f(\mathbf{x}_\delta) \in [0, 1]$) | 106.3 | **35.8** | 117.9 | **53.4** | 326.3 | **58.1** |

Table 1: The Fréchet Inception Distance (FID) score, measuring the quality of the generated explanations for the three prediction tasks. Lower FID corresponds to better image quality. Top (bottom) row corresponds the top (bottom) 10% of the decision interval.

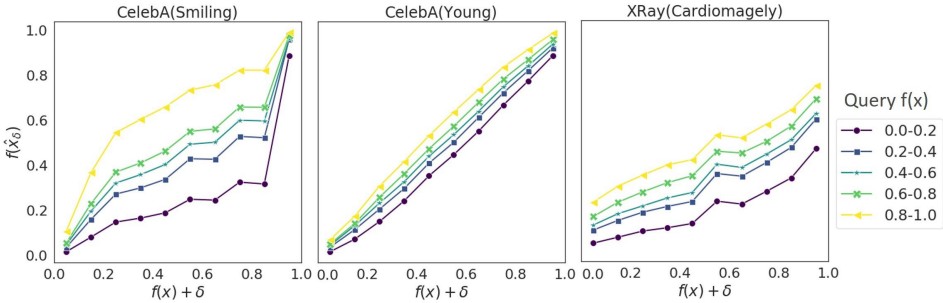

Figure 3: Plot of the expected outcome from the classifier, $f(\mathbf{x})+\delta$, against the actual response of the classifier on generated explanations, $f(\mathbf{x}_\delta)$. The monotonically increasing trend shows a positive correlation between $f(\mathbf{x}) + \delta$ and $f(\mathbf{x}_\delta)$, and thus the generated explanations are consistent with the expected condition.

**Identity preservation:** The generated explanations should differ only in semantic features associated with the target class, while retaining the identity of the query image. We extracted the latent embedding for real images ($E(\mathbf{x})$) and their corresponding explanations ($E(\mathbf{x}_\delta)$), for different values of $\delta$. We calculated **latent space closeness** as the percentage of the times, $\mathbf{x}_\delta$ is closest to the query image $\mathbf{x}$ as compared to other generated explanations

$$\forall \delta, \mathbf{x} \in \mathcal{X}, \quad ||E(\mathbf{x}) - E(\mathbf{x}_\delta)||_2 < \min_{\mathbf{m} \in \mathcal{I}_f(\mathcal{X} - \{\mathbf{x}\}, \delta)} ||E(\mathbf{x}_\delta) - E(\mathbf{m})||_2, \quad (10)$$

where, $\mathbf{m} \in \mathcal{I}_f(\mathcal{X} - \{\mathbf{x}\}, \delta))$ is the set of explanations generated for all the real images excluding the query image $\mathbf{x}$. Another, popular approach to quantify identity of two face images, is to perform **face verification**. We used state-of-the-art face recognition model trained on VGGFace2 dataset (Cao et al., 2018) as feature extractor for both real images and their corresponding fake explanations. For face verification, we calculated the closeness between real and fake image as cosine distance between their feature vectors. The faces were considered as verified *i.e.*, fake explanation have same identity as real image, if the distance is below 0.5. Table 2 summarizes the results.

Our method achieved high performance on localized attribute "smiling", which alters a relatively small region of the face image as compare to attribute "age" which affects the entire face. Medical images like chest x-ray have very fine grain details which are difficult to preserve in the generative process of GAN. Our explainer function preserves the high level features like shape and size of the lung, but it struggles to retain the low level features like anatomy of the breast and shape of the collar bones. Also, it should be noted that both the datasets have multiple images for same person, but we ignore this information in our analysis and treat each image as a different identity. We compared our performance against xGEM (Joshi et al., 2018). VAE explicitly minimizes for latent space closeness. The generated explanation by xGEM were blurry version of the query image. Hence, although they were close to query image in latent space, but they didn't preserve the identity of the individual as shown in face verification task and is evident in Figure 7 in appendix A.2. In comparison, our model achieved good performance on both the tasks.

## 4.2 COUNTERFACTUAL EVALUATION ON MEDICAL DATA

Cardiomegaly refers to an abnormal enlargement of the heart (Brakohiapa et al., 2017). To understand the explanations derived for Cardiomegaly target class, we overlaid the heart segmentation

|  | CelebA:Smiling | | CelebA:Young | | Xray:Cardiomegaly | |
|---|---|---|---|---|---|---|
|  | xGEM | Ours | xGEM | Ours | xGEM | Ours |
| **Latent Space Closeness** | **88.2** | 88.0 | **89.5** | 81.6 | 2.2 | **27.9** |
| **Face Verification Accuracy** | 0.0 | **85.3** | 0.0 | **72.2** | - | - |

Table 2: Identity preserving performance on three prediction tasks.

over the x-ray image and visualize the gradual change in heart size. The heart segmentation is shown as outlines in Figure 2, with their corresponding heart size (top values in parentheses). The heart segmentation is derive by training a UNet (Ronneberger et al., 2015) model on the segmentation in chest radiograph (SCR) dataset (van Ginneken et al., 2006). We registered $\mathbf{x}$ with its associated $\mathbf{x}_\delta$ and applied the resulting transformation to the heart masks of $\mathbf{x}$ to derive the heart masks for $\mathbf{x}_\delta$.

For population-level analysis, we plotted the average heart size of $\mathbf{x}_\delta$ vs the condition used for generation $(f(\mathbf{x}) + \delta)$ in Figure 4 (a). The plot shows a positive correlation between the heart size and the response of the classifier $f(\mathbf{x})$, which agrees with the definition of Cardiomegaly. To better understand the results, we divided the population into two groups, the first group $(\mathbf{x}^h; f(\mathbf{x}^h) < 0.1)$ consists of real images of healthy x-rays, and the second group $(\mathbf{x}^c; f(\mathbf{x}^c) > 0.9)$ contains real images of abnormal x-rays positive for Cardiomegaly. For $\mathbf{x}^h$ we generated counterfactual as $\mathbf{x}^c_\delta$ such that $f(\mathbf{x}^c_\delta) > 0.9$. Similarly, counterfactuals for $\mathbf{x}^c$ are derived as $\mathbf{x}^h_\delta$ such that $f(\mathbf{x}^h_\delta) < 0.1$. In Figure 4 (b), we show the distribution of heart size in the four groups. We reported the dependent t-test statistics for paired samples $(\mathbf{x}^h$ and $\mathbf{x}^c_\delta)$, $(\mathbf{x}^c$ and $\mathbf{x}^h_\delta)$. A significant p-value $\ll 0.001$ rejected the null hypothesis (*i.e.*, that the two groups have similar distributions). We also reported the independent two-sample t-test statistics for healthy $(\mathbf{x}^h$ and $\mathbf{x}^h_\delta$, p-value $> 0.01)$ and abnormal $(\mathbf{x}^c$ and $\mathbf{x}^c_\delta$, p-value $< 0.01)$ populations. Given higher p-values, we cannot reject the null hypothesis of identical average distributions with high confidence. Our model derived explanations successfully captured the change in heart size while generating counterfactual explanations.

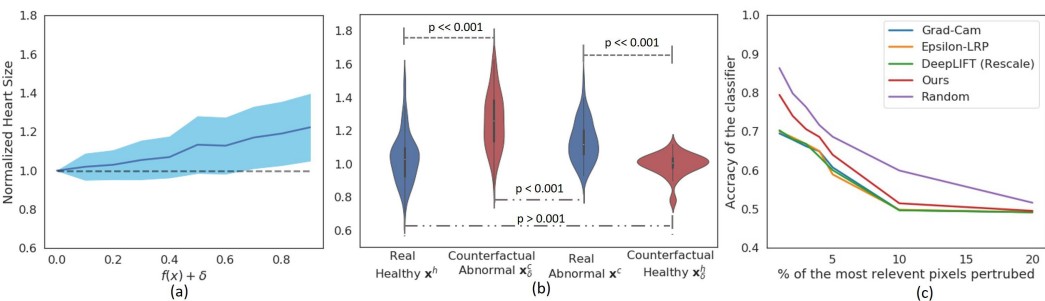

Figure 4: Cardiomegaly disease is associated with large heart size. In (a) we show the positive correlation between the heart size and the response of the classifier $f(\mathbf{x})$. (b) Comparison of the distribution of the heart size in the four groups. (c) Plot to show the drop in accuracy of the classifier as we perturb the most relevant pixels (relevance calculated from saliency map) in the image.

### 4.3 SALIENCY MAP

Saliency maps show the importance of each pixel of an image in the context of classification. Our method is not designed to produce saliency maps as a continuous score for every feature of the input. We extract an approximate saliency map by quantifying the regions that changed the most when comparing explanations at the opposing ends of the classification spectrum. For each query image, we generated two visual explanations corresponding to the two extremes of the decision boundary $(f(\mathbf{x}_\delta) = 0$ and $f(\mathbf{x}_\delta) = 1)$. The absolute difference between these explanations is our saliency map. Figure 5 shows the saliency map obtain from our method and its comparison with popular gradient based methods. We restricted the saliency maps obtained from different methods to have positive values and normalize them to range [0,1]. Subjective, the saliency maps produced by our method are very localized and are comparable to the other methods.

We adapted the metric introduced in (Samek et al., 2016) to compare the different saliency maps. In an iterative procedure, we progressively replace a percentage of the most relevant pixels in an image (as given by the saliency map) with random values sampled from a uniform distribution. We observe the corresponding change in the classification performance as shown in Figure 4 (c). All the methods experienced a drop in the accuracy of the classifier with increase in the fraction of perturb pixels. The saliency maps produced by our model is significantly better than random maps and are comparable to the other saliency map methods. It should be noted that, there are many ways to quantify important regions in a image, using the series of explanations generated by our method. We didn't optimize to find the best saliency map and showed results for one such method.

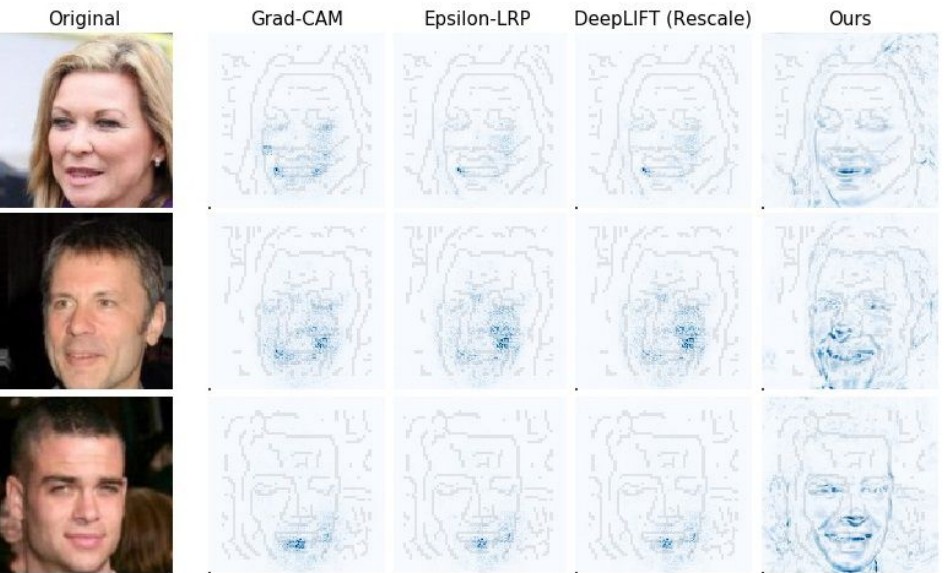

Figure 5: Our comparison with popular gradient-based saliency map producing methods on the prediction task of identifying smiling faces in CelebA dataset.

## 4.4 BIAS DETECTION

Our model can discover confounding bias in the data used for training the black-box classifier. Confounding bias provides an alternative explanation for an association between the data and the target label. For example, a classifier trained to predict the presence of a disease may make decisions based on hidden attributes like gender, race, or age. In a simulated experiment, we trained two classifiers to identify smiling vs not-smiling images in the CelebA dataset. The first classifier $f_{\text{Biased}}$ is trained on a biased dataset, confounded with gender such that all smiling images are of male faces. We train a second classifier $f_{\text{No-biased}}$ on an unbiased dataset, with data uniformly distributed with respect to gender. Note that we evaluate both the classifiers on the same validation set. Additionally, we assume access to a proxy Oracle classifier $f_{\text{Gender}}$ that perfectly classifies the confounding attribute $i.e.$, gender. As shown in Cohen et al. (2018), if the training data for the GAN is biased, then the inference would reflect that bias. In Figure 6, we compare the explanations generated for the two classifiers. The visual explanations for the biased classifier change gender as it increases the amount of smile. We adapted the confounding metric proposed in Joshi et al. (2018) to summarize our results in Table 3. Given the data $\mathcal{D} = \{(\mathbf{x}_i, y_i, a_i), \mathbf{x}_i \in \mathcal{X}, y_i, a_i \in \mathcal{Y}\}$, we quantify that a classifier is confounded by an attribute $a$ if the generated explanation $\hat{x}_\delta$ has a different attribute $a$, as compared to query image $\mathbf{x}$, when processed through the Oracle classifier $f_{\text{Gender}}$. The metric is formally defined as $\mathbb{E}_{\mathcal{D}}[1(g^*(\mathbf{x}_\delta) \neq a)]/|\mathcal{D}|$. For a biased classifier, the Oracle function predicted the female class for the majority of the images, while the unbiased classifier is consistent with the true distribution of the validation set for gender. Thus, we the fraction of generated explanations that changed the confounding attribute "gender' was found to be high for the biased classifier.

| Black-box classifier | Target Label | |
| --- | --- | --- |
| | Smiling | Not-Smiling |
| $f_{\text{Biased}}$ | Male: 0.52 | Male: 0.18 |
| | Female: 0.48 | Female: 0.82 |
| | Overall: **0.12** | Overall: **0.35** |
| $f_{\text{No-biased}}$ | Male: 0.48 | Male: 0.47 |
| | Female: 0.52 | Female: 0.53 |
| | Overall: 0.07 | Overall: 0.08 |

Table 3: Confounding metric for biased detection. For target label "Smiling" and "Not-Smiling", the explanations are generated using condition $f(x) + \delta > 0.9$ and $f(x) + \delta < 0.1$ respectively. The Male and Female values quantifies the fraction of the generated explanations classifier as male or female, respectively by oracle classifier $f_{\text{Gender}}$. The overall value quantifies the fraction of the generated explanations who have different gender as compared to the query image. A small overall value shows least bias.

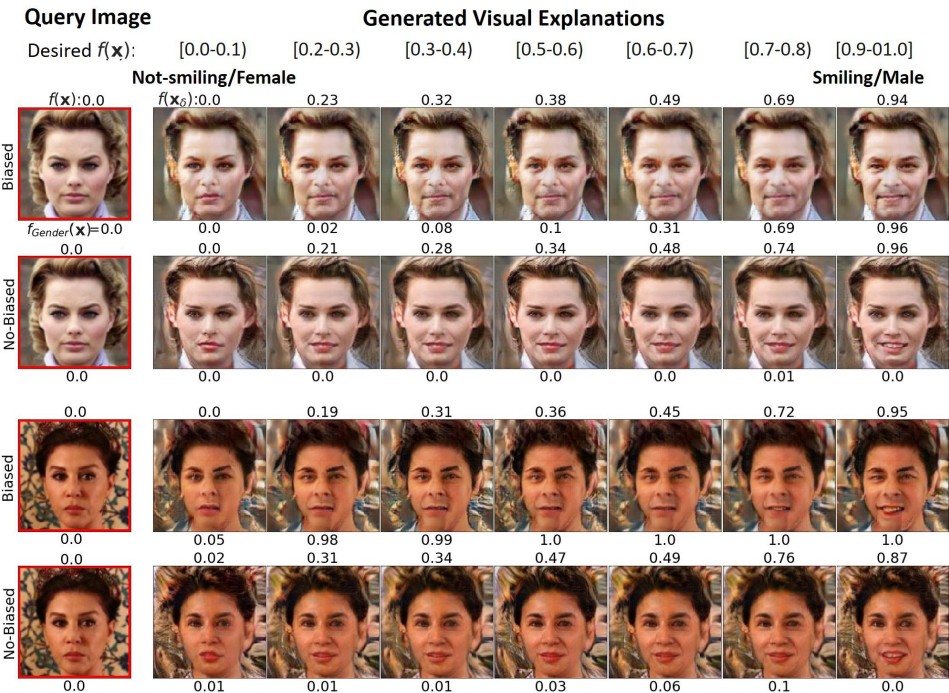

Figure 6: The visual explanations for two classifiers, both trained to classify "Smiling" attribute on CelebA dataset. For each example, the top row shows results from "Biased" classifier whose data distribution is confounded with "Gender". The bottom row shows explanations from "No-Biased" classifier with uniform data distribution w.r.t gender. The top label indicates output of the classifier and the bottom label is the output of an oracle classifier for the con-founding attribute gender. The visual explanations for the "Biased" classifier changes the gender as it adds smile on the face.

## 5 CONCLUSION

In this paper, we proposed a novel interpretation method that explains the decision of a black-box classifier by producing natural-looking, gradual perturbations of the query image, resulting in an equivalent change in the output of the classifier. We evaluated our model on two very different datasets, including a medical imaging dataset. Our model produces high-quality explanations while preserving the identity of the query image. Our analysis shows that our explanations are consistent with the definition of the target disease without explicitly using that information. Our method can also be used to generate a saliency map in a model agnostic setting. In addition to the interpretability advantages, our proposed method can also identify plausible confounding biases in a classifier.

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

## A    APPENDIX

### A.1    IMPLEMENTATION DETAILS

The architecture for the generator and discriminator is adapted from Miyato & Koyama (2018). The image encoding learned by encoder $E(\mathbf{x})$ is fed into the generator. The condition $c_f(\mathbf{x}, \delta)$ is passed to each resnet block in the generator, using conditional batch normalization. The generator has five resnet blocks, where each block consists of BN-ReLU-Conv3-BN-ReLU-Conv3. BN is batch normalization, ReLU is activation function, and Conv3 is the convolution filter. The encoder function uses the same structure but downsamples the image. The discriminator function has five resnet blocks, each of which has the form ReLU-Conv3-ReLU-Conv3.

## A.2 xGEM IMPLEMENTATION

We refers to Joshi et al. (2019) for the implementation of xGEM. First, VAE is trained to generate face images. The VAE used is available at:https://github.com/LynnHo/VAE-Tensorflow. All settings and architectures were set to default values. The original code generates an image of dimension 64x64. We extended the given network to produce an image with dimensions 128x128. The pre-trained VAE is then extended to incorporate the cross-entropy loss for flipping the label of the query image. The model evaluates the cross-entropy loss by passing the generated image through the classifier. Figure 7 shows the qualitative difference between the explanations generated by our proposed method and xGEM.

## A.3 EXTENDED RESULTS FOR EVALUATING THE CRITERIA OF THE EXPLAINER

Here, we provide results for four more prediction tasks on celebA dataset: no-beard or beard, heavy makeup or light makeup, black hair or not back hair, and bangs or no-bangs. Figure 8 shows the qualitative results, an extended version of results in Figure 2. We evaluated the results from these prediction tasks for compatibility with black-box $f$ (*see* Figure 9), data consistency and self consistency (*see* Table 4).

| Prediction Task | Data Consistency (FID) | | | Self Consistency | |
|---|---|---|---|---|---|
| | Present | Absent | Overall | LSC | FVA |
| Smiling vs Not-smiling | 46.9 | 56.3 | 35.8 | 88.0 | 85.3 |
| Young vs Old | 67.5 | 74.4 | 53.4 | 81.6 | 72.2 |
| No beard vs Beard | 79.2 | 72.3 | 45.4 | 89.6 | 83.3 |
| Heavy makeup vs Light makeup | 64.9 | 98.2 | 39.2 | 89.2 | 75.3 |
| Black hair vs Not black hair | 55.8 | 72.8 | 34.8 | 79.4 | 81.6 |
| Bangs vs No bangs | 54.1 | 57.8 | 40.6 | 76.5 | 87.3 |

Table 4: Our model results for six prediction tasks on CelebA dataset. FID (Fréchet Inception Distance) score measures the quality of the generated explanations. Lower FID is better. LSC (Latent Space Closeness) quantifies the fraction of the population where generated explanation is nearest to the query image than any other generated explanation in embedding space. FVA (Face verification accuracy) measures percentage of the times the query image and generated explanation have same face identity as per model trained on VGGFace2. Higher LSC and FVA is better.

## A.4 HUMAN EVALUATION

We used Amazon Mechanical Turk (AMT) to conduct human experiments to demonstrate that the progressive exaggeration produced by our model is visually perceivable to humans. We presented AMT workers with three tasks. In the first task, we evaluated if humans can detect the relative order between two explanations produced for a given image. We ask the AMT workers, "Given two images of the same person, in which image is the person younger (or smiling more)?" (*see* Figure 10). We experimented with 200 query images and generated two pairs of explanations for each query image (*i.e.*, 400 hits). The first pair (*easy*) imposed the two images are samples from opposite ends of the explanation spectrum (counterfactuals), while the second pair (*hard*) makes no such assumption.

In the second task, we evaluated if humans can identify the target class for which our model has provided the explanations. We ask the AMT workers, "What is changing in the images? (age, smile, hair-style or beard)". We experimented with 100 query images from each of the four attributes (*i.e.*, 400 hits). In the third task, we demonstrate that our model can help the user to identify problems like possible bias in the black-box training. Here, we used the same setting as in the second task but also showed explanations generated for a biased classifier. We ask the AMT workers, "What is changing in the images? (smile or smile and gender)" (*see* Figure 10). We generated explanations for 200 query images each, from a biased-classifier ($f_{\text{Biased}}$) explainer from Section 4.4 and an unbiased classifier ($f_{\text{No-biased}}$) explainer (*i.e.*, 400 hits). In all the three tasks, we collected eight votes for each task, evaluated against the ground truth, and used the majority vote for calculating accuracy.

We summarize our results in Table 5. In the first task, the annotators achieved high accuracy for the *easy* pair when there was a significant difference among the two explanation images, as compared to the *hard* pair when the two explanations can have very subtle differences. Overall, the annotators were successful in identifying the relative order between the two explanation images.

In the second task, the annotators were generally successful in correctly identifying the target class. The target class "bangs" proved to be the most difficult to identify, which was expected. The generated images for "bangs" were qualitatively, the most subtle. For the third task, the correct answer was always the target class *i.e.*, "smile". In the case of biased classifier explainer, the annotators selected "Smile and Gender" 12.5% of the times. The gradual progression made by the explainer for a biased classifier was very subtle and was changing large regions of the face as compared to the unbiased explainer. The difference is much more visible when we compare the explanation generated for the same query image for a biased and no-biased classifier, as in Figure 6. But in a realistic scenario, the no-biased classifier would not be available to compare against. Nevertheless, the annotators detected bias at roughly the same level of accuracy as our classifier (Table 3). Future work could improve upon bias detection.

| Annotation Task | Overall | | Sub categories | | |
|---|---|---|---|---|---|
| | Accuracy | $\kappa$-statistic | Category | Accuracy | $\kappa$-statistic |
| **Task-1 (Age)** | 83.5% | 0.41 (Moderate) | Hard | 73% | 0.31 (Fair) |
| | | | Easy | **94%** | 0.51 (Moderate) |
| **Task-1 (Smile)** | 77.5% | 0.28 (Fair) | Hard | 66% | 0.23 (Fair) |
| | | | Easy | **89.5%** | 0.32 (Fair) |
| **Task-2 (Identify Target Class)** | 77% | 0.35 (Fair) | Age | 72% | - |
| | | | Smile | **99%** | - |
| | | | Bangs | 50% | - |
| | | | Beard | 87% | - |
| **Task-3 (Bias Detection)** | 93.75% | 0.14 (Slight) | $f_{\text{Biased}}$ | 87.5% | 0.09 (Slight) |
| | | | $f_{\text{No-biased}}$ | **100%** | 0.02 (Slight) |

Table 5: Summarizing the results of human evaluation. The $\kappa$-statistics measure inter-rater agreement for qualitative classification of items into some mutually exclusive categories. One possible interpretation of $\kappa$ as given in Viera et al. (2005) is $< 0.0$: Poor, $0.01 - 0.2$: Slight, $0.21 - 0.40$: Fair, $0.41 - 0.60$: Moderate, $0.61 - 0.80$: Substantial and $0.81 - 1.00$: Almost perfect agreement.

## A.5 EVALUATING CLASS DISCRIMINATION

In multi-label settings, multiple labels can be true for a given image. In this test, we evaluated the sensitivity of our generated explanations to the class being explained. We consider a classifier trained to identify multiple attributes: young, smiling, black-hair, no-beard and bangs in face images from CelebA dataset. We used our model to generate explanations while considering one of the attributes as the target. Ideally, an explanation model trained to explain a target attribute should produce explanations consistent with the query image on all the attributes beside the target. Figure 11 plots the fraction of the generated explanations, that have flipped in source attribute as compared to the query image. Each column represents one source attribute. Each row is one run of our method to explain a given target attribute.

## A.6 ABLATION STUDY

Our proposed model has three types of loss functions: adversarial loss from cGAN, KL loss, and reconstruction loss. The three losses enforce the three properties of our proposed explainer function: data consistency, compatibility with $f$, and self-consistency, respectively. In the ablation study, we quantify the importance of each of these components by training different models, which differ in one hyper-parameter while rest are equivalent ($\lambda_{\text{cGAN}} = 1$, $\lambda_f = 1$ and $\lambda_{\text{rec}} = 100$). For **data consistency**, we evaluate Fréchet Inception Distance (FID). FID score measures the visual quality of the generated explanations by comparing them with the real images. We show results for two

groups. In the first group, we consider real and fake images where the classifier has high confidence in *presence* of the target label *i.e.*, $f(\mathbf{x}_\delta), f(\mathbf{x}) \in [0.9, 1.0]$. In second group, the target label is *absent i.e.*, $f(\mathbf{x}_\delta), f(\mathbf{x}) \in [0.0, 0.1]$. We also report an overall score by considering all the real and generated explanations together. For **compatability with** $f$ we plotted the desired output of the classifier *i.e.*, $f(\mathbf{x}) + \delta$ against the actual output of the classifier $f(\mathbf{x}_\delta)$ for the generated explanations. For **self consistency**, we calculated the Latent Space Closeness (LSC) measure and Face verification accuracy (FVA). LSC quantifies the fraction of the population in which the generated explanation is nearest to the query image than any other generated explanation in embedding space. FVA measures the percentage of the instances in which the query image and generated explanation have the same face identity as per the model trained on VGGFace2. For the ablation study, we consider the prediction task of young vs old on the CelebA dataset. Figure 12 shows the results for compatibility with $f$. Table 6 summarizes the results for data consistency and self-consistency.

| Configuration | | | Data Consistency (FID) | | | Self Consistency | |
|---|---|---|---|---|---|---|---|
| $\lambda_{cGAN}$ | $\lambda_f$ | $\lambda_{rec}$ | Present | Absent | Overall | LSC | FVA |
| 0 | 1 | 100 | 69.7 | 105.7 | 67.2 | 96.1 | 99.8 |
| 1 | 1 | 100 | **67.5** | **74.4** | **53.4** | 81.6 | 72.2 |
| 10 | 1 | 100 | 89.4 | 105.2 | 63.0 | 68.0 | 82.7 |
| 100 | 1 | 100 | 71.6 | 80.6 | 44.26 | 75.3 | 18.0 |
| 1 | 0 | 100 | 66.2 | 66.2 | 44.9 | 77.2 | 99.4 |
| 1 | 1 | 100 | 67.5 | 74.4 | 53.4 | 81.6 | 72.2 |
| 1 | 10 | 100 | 95.5 | 90.4 | 62.4 | 71.83 | 96.8 |
| 1 | 100 | 100 | 77.4 | 73.1 | 71.2 | 55.4 | 42.23 |
| 1 | 1 | 0 | 116.2 | 118.9 | 72.2 | 16.6 | 0.0 |
| 1 | 1 | 1 | 63.0 | 78.6 | 61.6 | 32.2 | 5.5 |
| 1 | 1 | 10 | 87.6 | 83.6 | 65.7 | 71.5 | **88.8** |
| 1 | 1 | 100 | 67.5 | 74.4 | 53.4 | **81.6** | 72.2 |

Table 6: Our model with ablation on prediction task of young vs old on CelebA dataset. FID (Fréchet Inception Distance) score measures the quality of the generated explanations. Lower FID is better. LSC (Latent Space Closeness) quantifies the fraction of the population where generated explanation is nearest to the query image than any other generated explanation in embedding space. FVA (Face verification accuracy) measures percentage of the times the query image and generated explanation have same face identity as per model trained on VGGFace2. Higher LSC and FVA is better.

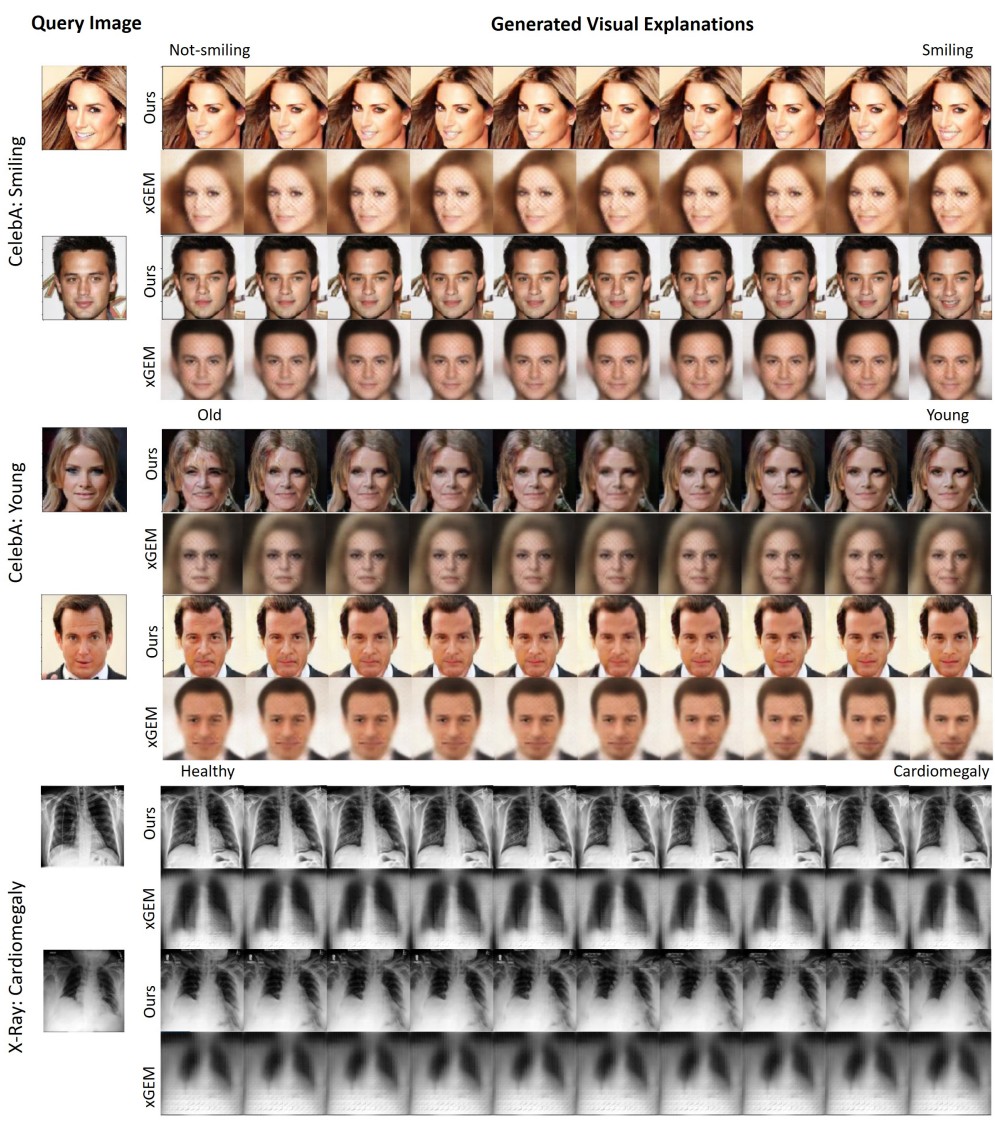

Figure 7: Visual explanations generated for three prediction tasks on CelebA dataset. The first column shows the query image, followed by the corresponding generated explanations.

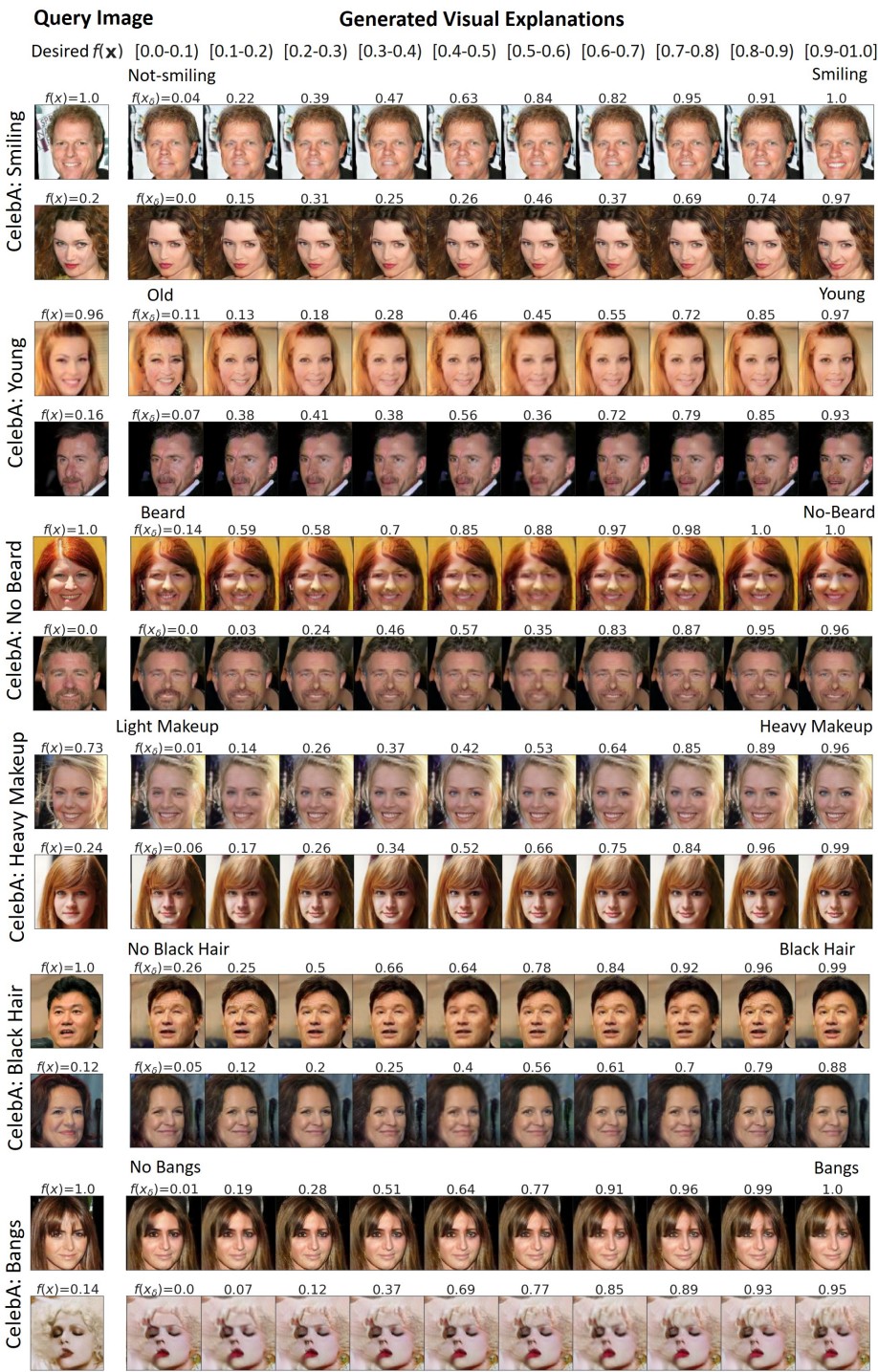

Figure 8: Visual explanations generated for six prediction tasks on CelebA dataset. The first column shows the query image, followed by the corresponding generated explanations. The values above each image are the output of the classifier $f$.

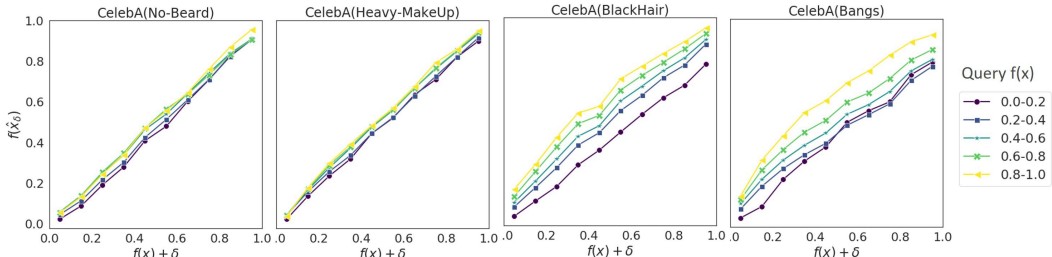

Figure 9: Plot of the expected outcome from the classifier, $f(\mathbf{x}) + \delta$, against the actual response of the classifier on generated explanations, $f(\mathbf{x}_\delta)$. The monotonically increasing trend shows a positive correlation between $f(\mathbf{x}) + \delta$ and $f(\mathbf{x}_\delta)$, and thus the generated explanations are consistent with the expected condition.

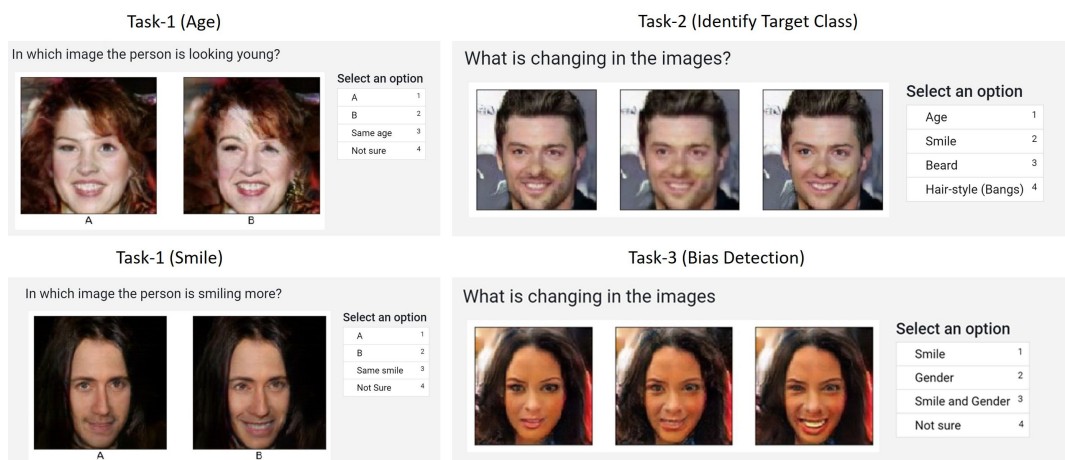

Figure 10: The interface for the human evaluation done using Amazon Mechanical Turk (AMT). Task-1 evaluated if humans can detect the relative order between two explanations. Task-2 evaluated if humans can identify the target class for which our model has provided the explanations. Task-3 demonstrated that our model can help the user to identify problems like possible bias in the black-box training.

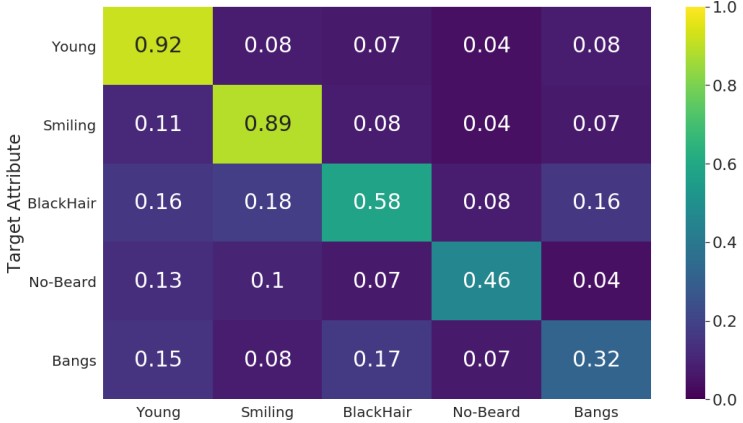

Figure 11: Each cell is the fraction of the generated explanations, that have flipped in source attribute as compared to the query image. The x-axis is source attribute and y-axis is the target attribute for which explanation is generated. Note: This is not a confusion matrix.

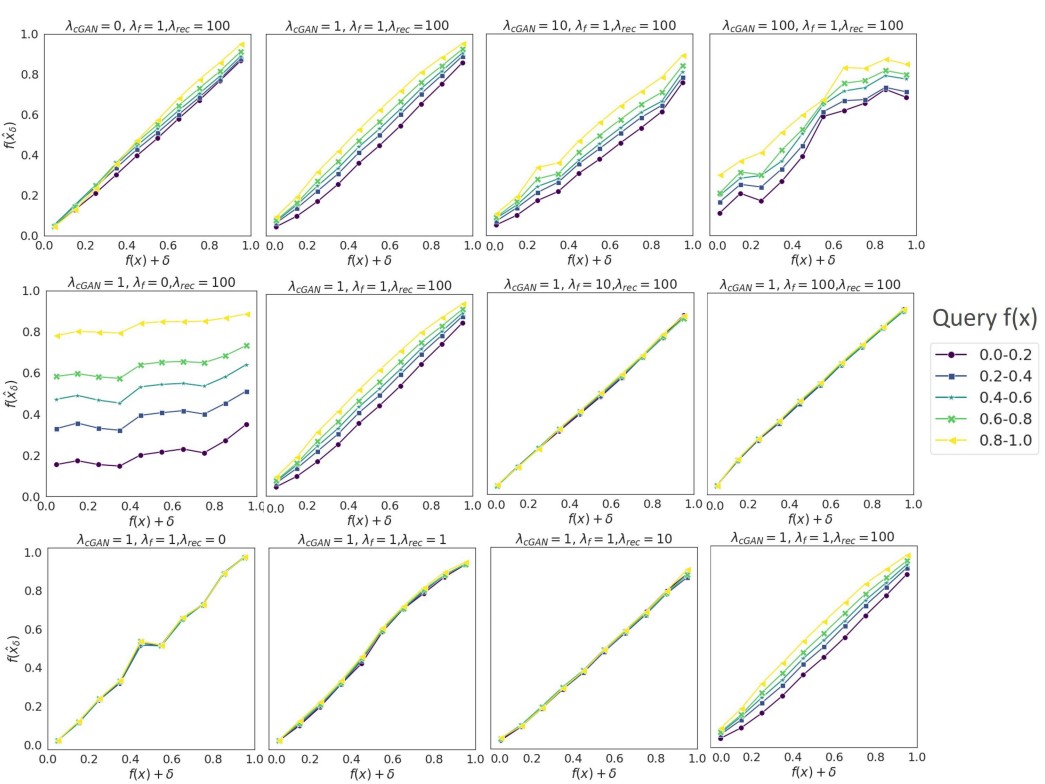

Figure 12: Ablation study to show the effect of KL loss term. Plot of the expected outcome from the classifier, $f(\mathbf{x}) + \delta$, against the actual response of the classifier on generated explanations, $f(\mathbf{x}_\delta)$.

