# OpenReview forum: "Explanation  by Progressive  Exaggeration"
_ICLR.cc/2020/Conference — Accept (Spotlight)_

### Official Review · AnonReviewer1 · 2019-10-22
**Official Blind Review #1**

**Rating:** 8

**Review:**

Here are the claims I could find in the intro:
"Given a query input to a black-box, we aim at explaining the outcome by providing plausible and progressive variations to the query that can result in a change to the output"
 > This is well supported as the model generates these and it is very reasonable that it can.
"the counterfactually generated samples are realistic-looking"
> The images seem to support this.
"the method can be used to detect bias in training of the predictor"
> Section 4.4 makes it really clear that, at least in the described setting, it works.

I think the idea could be presented in a better way. The general concept of exaggerating a feature that represented a class seems novel and exciting. Just based on the novelty of that alone I think this is worth accepting. I would imagine there would be a cleaner way of achieving all this but maybe it is all necessary.

I don't understand Figure 1a. I don't think this helps to illustrate the point. M_z seems to just be a bottleneck but the writing makes it seem like it is more.

Section 4.2 is a bit hard to read. It is not clear for me what is the goal of this section.

Section 4.4 seems very similar to the idea in this work https://arxiv.org/abs/1805.08841 which studied how bias in CycleGANs can be seen when you vary the bias which I think should be cited here.

Typos:
"our application of interested"


**Experience Assessment:**

I have published in this field for several years.

**Review Assessment: Checking Correctness Of Derivations And Theory:**

I assessed the sensibility of the derivations and theory.

**Review Assessment: Checking Correctness Of Experiments:**

I assessed the sensibility of the experiments.

**Review Assessment: Thoroughness In Paper Reading:**

I read the paper at least twice and used my best judgement in assessing the paper.

---

> ### Author Response · Authors · 2019-11-09
> **Reply to Official Blind Review #1**
>
> Thank you for your valuable and constructive comments.
>
> 1. I don't understand Figure 1a. I don't think this helps to illustrate the point. M_z seems to just be a bottleneck but the writing makes it seem like it is more.
>
> [Ans] We agree that M_z is just a bottleneck. Through Figure 1a, we want to show the abstraction of the changes (perturbations) in the bottleneck.  We hope the figure is conveying the message. We have added additional text to explain the figure better. If it is still misleading, we can remove the figure.
>
> Figure 1a is showing that the perturbation of the image happens not in high dimensional image space (M_x) but in low dimensional embedding space i.e. M_z. And there is a correspondence between image space (M_x) and latent space (M_z).
>
> Also, Figure 1a is explaining the meaning of the desired perturbation (\delta). Our proposed explainer function takes two arguments: a query image and the desired perturbation. Figure 1a demonstrates that the desired perturbation is the desired change in f(x). Most of the earlier work denoted \delta as the amount of change in the input image (x).
>
>
> 2. Section 4.2 is a bit hard to read. It is not clear for me what is the goal of this section.
>
> [Ans] We understand that our audience will be less familiar with the x-ray images. We are showing the results of our model for evaluating Cardiomegaly disease on chest x-ray. Cardiomegaly means an enlarged heart. We overlaid the heart segmentation over the x-ray image, to help the readers in visualizing the gradual change in heart size. The black-box model didn’t use the heart segmentation or the heart size information to classify chest x-ray as Cardiomegaly.  Our explanation model was successful in exaggerating the correct features (increasing the heart size), which comply with the clinical definition of the disease.
>
>
> In Section 4.2, we showed that on population-level, when we generate a counterfactual (abnormal) for a normal chest x-ray, it has a higher heart size as compared to the normal population and vice-versa. Thus, the explainer was successful in correlating the heart size with the disease. Hence, the explainer verified that the back-box is considering the correct feature (heart-size) for identifying the target class (cardiomegaly).
>
> 3. Section 4.4 seems very similar to the idea in this work https://arxiv.org/abs/1805.08841 which studied how bias in CycleGANs can be seen when you vary the bias which I think should be cited here.
>
> [Ans] The conclusion of the suggested work (Distribution Matching Losses Can Hallucinate Features in Medical Image Translation) is that if the training data has biased, then GAN inference will reflect that bias. Section 4.4 is using this conclusion to detect bias. Thank you for introducing this relevant work. We have cited this paper in our updated version
>
>
> 4. Typos: "our application of interested"
> [Ans] We have corrected the typo in our updated version.
>
> We will soon upload our revised version.

---

### Official Review · AnonReviewer3 · 2019-10-29
**Official Blind Review #3**

**Rating:** 6

**Review:**

The paper presents a method for explaining the output of black box classification of images.  The method generates  gradual perturbation of outputs in response to gradually perturbed input queries. The rationale is that, by looking at these, humans can interpret the classification mechanics.

The presentation is clear. The coverage of prior work is sufficient (although references should point to the published work instead of arxiv entries, when the former is available).

One question that is not addressed is how efficient is this method, in terms of computational cost. This is a method that increases the amount of input data (through perturbation). What is the minimum amount of input data that needs to be perturbed in this way, before the method can become human interpretable?

Also, ideally any work on human interpretability of ML should be evaluated on humans. If not, it is an approximation, and it should be presented and reasoned as such (with a discussion of limitations and caveats, for instance).

**Experience Assessment:**

I have read many papers in this area.

**Review Assessment: Checking Correctness Of Derivations And Theory:**

I did not assess the derivations or theory.

**Review Assessment: Checking Correctness Of Experiments:**

I assessed the sensibility of the experiments.

**Review Assessment: Thoroughness In Paper Reading:**

I made a quick assessment of this paper.

---

> ### Author Response · Authors · 2019-11-09
> **Reply to Official Blind Review #3**
>
> Thank you reviewer for your valuable and constructive comments.
>
> 1. The presentation is clear. The coverage of prior work is sufficient (although references should point to the published work instead of arxiv entries, when the former is available).
>
> [Ans] We have updated the references in the paper to the published work.
>
> 2. One question that is not addressed is how efficient is this method, in terms of computational cost. This is a method that increases the amount of input data (through perturbation). What is the minimum amount of input data that needs to be perturbed in this way, before the method can become human interpretable?
>
> [Ans]
> We want to answer this question in terms of computational and statistical efficiency. Computationally, our model is very efficient. At inference time, for a new image, only a single forward pass is required to generate a series of perturbation images (explanation).
>
> In terms of statistical efficiency, yes, our model requires a minimum amount of input data for GAN training. The training data should be sufficient for training GAN and for producing realistic-looking results. However, the end-user can use any data that is compatible with the black-box model to train the explainer function, and not necessarily the same data that was used to train the black-box model. Also, we don’t require labeled (supervised) data for training our explainer function.
>
> 3. Also, ideally any work on human interpretability of ML should be evaluated on humans. If not, it is an approximation, and it should be presented and reasoned as such (with a discussion of limitations and caveats, for instance).
>
> [Ans] We are currently running human experiments to test our model and will add the results in the revision. For human evaluation, we are running three tasks on Amazon Mechanical Turk.
>
> In the first task, we demonstrate how humans perceive “the progressive exaggeration” aspect of our explanation. In this task, we showed users two explanations create by our model for the same individual and ask the user to compare them in terms of a target class like age and smile (e.g. Identify the image in which the person is smiling more?).
>
> In the second task, we show that our explanations help the user to understand better, the target class for the classifier. In this task, we showed users a series of images with gradual exaggeration of a target class, and ask the users to identify the target class. (e.g what is changing in the below images? Option: Age, Smile, Gender, Nothing, Something else).
>
> In the third task, we demonstrate that our model helped the user to identify problems (biased training) in a black-box. Here, we used the same setting as in the second task but also showed explanations generated from a biased classifier.
>
>
> We will soon upload our revised version.

---

### Author Response · Authors · 2019-11-15
**Summary of the updates to the paper.**

We want to thank the reviewers for their valuable and constructive feedback.

We have made the following changes in the revision of our paper.
1. Updated references to point to published articles.
2. Added human evaluation experiment in Appendix section A.4. We used Amazon Mechanical Turk (AMT) to conduct human experiments to demonstrate that the progressive exaggeration produced by our model is visually perceivable to humans.
3. Updated the description of Figure 1 to better explain the intent of the schematic figure.
4. Updated Section 4.2 to better explain the experiment with medical data.
5. Updated Section 4.1 with results from xGEM for data consistency and identity preservation goals.
6. Added another evaluation metric for identity preservation in Section 4.1. The new metric is based on face verification.
7. Added Appendix section with further experiments to demonstrate our model's compatibility with a multi-label classifier and, an ablation study, to show the relative importance of each of the three terms in the final loss function in equation 9.


We hope that these changes address the reviewers' concerns. We are happy to provide any more details. We will also release our code on GitHub very soon.

Regards,
The Authors

---

### Decision · Program_Chairs · 2019-12-19

**Decision:**

Accept (Spotlight)

**Comment:**

This paper presents an idea for interpolating between two points in the decision-space of a black-box classifier in the image-space, while producing plausible images along the interpolation path. The presentation is clear and the experiments support the premise of the model.
While the proposed technique can be used to help understanding how a classifier works, I have strong reservations in calling the generated samples "explanations". In particular, there is no reason for the true explanation of how the classifier works to lie in the manifold of plausible images. This constraint is more of a feature to please humans rather than to explain the geometry of the decision boundary.
I believe this paper will be well-received and I suggested acceptance, but I believe it will be of limited usefulness for robust understanding of the decision boundary of classifiers.